# A Randomized Comparison of Non-Channeled Glidescope^TM^ Titanium Versus Channeled KingVision^TM^ Videolaryngoscope for Orotracheal Intubation in Obese Patients with BMI > 35 kg·m^−2^

**DOI:** 10.3390/diagnostics10121024

**Published:** 2020-11-29

**Authors:** Tomas Brozek, Jan Bruthans, Michal Porizka, Jan Blaha, Jitka Ulrichova, Pavel Michalek

**Affiliations:** 1Department of Anaesthesiology and Intensive Medicine, General University Hospital and First Medical Faculty of the Charles University, 128 00 Prague, Czech Republic; tomas.brozek@vfn.cz (T.B.); jan.bruthans@vfn.cz (J.B.); michal.porizka@vfn.cz (M.P.); jan.blaha@vfn.cz (J.B.); jitka.ulrichova@vfn.cz (J.U.); 2Medical Faculty, Masaryk University, 625 00 Brno, Czech Republic; 3Department of Anaesthesia, Antrim Area Hospital, Antrim BT41 2RL, UK

**Keywords:** obesity, videolaryngoscopy, King Vision^TM^ laryngoscope, Glidescope Titanium^TM^ laryngoscope, non-channeled blade, channeled blade

## Abstract

Videolaryngoscopes may improve intubating conditions in obese patients. A total of 110 patients with a body mass index > 35 kg∙m^−2^ were prospectively randomized to tracheal intubation using non-channeled Glidescope Titanium or channeled King Vision videolaryngoscope. The primary outcome was the time to tracheal intubation. Secondary outcomes included: total success rate, number of attempts, the quality of visualization, peri-procedural and post-proceduralcomplications. Time to the first effective breath was shorter with the King Vision (median; 95% CI)—36; 34–39 s vs. 42; 40–50 in the Glidescope group (*p* = 0.007). The total success rate was higher in the Glidescope group—100% vs. 89.1% (*p* = 0.03). There was a higher incidence of moderate and difficult laryngoscopy in the King Vision group. No difference was recorded in first attempt success rates, total number of attempts, use of additional maneuvers, intraoperative trauma, or any significant decrease in SpO_2_ during intubation. No serious complications were noted and the incidence of postoperative complaints was without difference. Although tracheal intubation with King Vision showed shorter time to the first breath, total success was higher in the Glidescope group, and all but one patients where intubation failed with the KingVision were subsequently intubated with the Glidescope.

## 1. Introduction

Obesity has become a significant worldwide healthcare problem. The World Health Organization defines obesity as Body Mass Index (BMI) value greater than 30 kg∙m^−2^, with the class I obesity between 30–35 kg∙m^−2^, class II obesity between 35–40 kg∙m^−2^, and class III obesity more than 40 kg∙m^−2^. Overall, the total number of obese people has almost tripled since the 1970s. In the United States, the prevalence is 42.4% among the adult population, without any significant difference between males and females [1]. The prevalence of obesity in European countries is lower at 15.9%; however, the combined prevalence of overweight and obese patients is 44.7% in men and 30.5% in women [2]. 

Obesity is associated with increased morbidity, including cancer, cardiovascular, and metabolic diseases [3]. Airway management including mask ventilation, laryngoscopy, tracheal intubation, and extubation is generally more difficult in class II and class III obesity than in lean patients due to the accumulation of fatty tissue inside the pharynx, tongue, neck, and chest [4,5]. 

The optimal technique of tracheal intubation in the obese population has not yet been definitely determined. Several randomized controlled studies [6,7,8,9] have suggested that videolaryngoscopy may provide better visualization of the vocal cords and improved overall intubating conditions in comparison with Macintosh direct laryngoscopy. A subsequent meta-analysis concluded that videolaryngoscopy provided a better success rate, faster intubation time, and improved visualization of the glottis than the Macintosh laryngoscope [10]. However, channeled videolaryngoscopes have not yet been compared with non-channeled devices in obese patients. Therefore we designed and carried out a single-center randomized trial to compare intubation times between the videolaryngoscopic non-channeled blade Glidescope Titanium® (Verathon Inc., Seattle, WA, USA) (Figure 1) with the disposable channeled blade of the King Vision® (Ambu Ltd., Copenhagen, Denmark) videolaryngoscope (Figure 2). We also evaluated secondary outcomes including the complex intraoperative performance of both videolaryngoscopes, and differences in postoperative complications.

## 2. Materials and Methods 

### 2.1. Study Design

This single-center, prospective randomized study was approved by the Ethics Committee of General University Hospital (No. 826/16 S-IV, approved on 19 May 2016, chairperson: dr. Josef Sedivy). The study was registered on www.anzctr.org.au (ACTRN12616001493437, principal investigator: Pavel Michalek) on 27 October 2016. Patients were randomized from 4 January 2017, till 21 May 2018. This manuscript was prepared in concordance with the following guidelines: CONSORT 2010 [11] and its 2012 extension [12]. The complete version of the study protocol is available by contacting the corresponding author. 

### 2.2. Study Participants

Adult patients (age 18–90 years) of both genders, American Society of Anesthesiologists (ASA) physical status I-III scheduled for an elective procedure of general surgery, gynecology, urology, maxillofacial surgery, ENT, and orthopedics requiring tracheal intubation were invited to participate in the study. The main inclusion criterion was Body Mass Index (BMI) > 35 kg∙m^−2^ while exclusion criteria were age less than 18 years or more than 90 years, mouth opening less than 2 cm, history of difficult airway requiring flexible fiberoptic intubation, emergency surgery, increased risk for gastric content regurgitation or aspiration, and inability to communicate in the Czech language. All participants received the participant information sheet at least 24 h before the procedure and signed the informed consent before randomization. Potential participants were identified through the booking list and then contacted during the pre-assessment clinic. The study coordinators described the design of the trial, responded to questions, provided participant information sheet, and informed consent form. The informed consent form was signed the evening before surgery and the attending anesthesiologist performed preoperative airway assessment. 

### 2.3. Investigators, Their Preparation and Training 

Five investigators performed all tracheal intubations of the study patients. They were all board-certified anesthesiologists, with at least 8 years of work experience. None of the study devices had been a routine intubation tool in our institution before the start of the trial. The investigators underwent complex training before the initiation of the study. The training included a video demonstration of both devices, intubating manikins with both devices, and at least 20 intubations on non-study patients with each videolaryngoscope prior to enrollment of the first patient. 

### 2.4. Randomization and Blinding

A sequence of computer-generated random numbers was created using a randomization software (GraphPad Software, La Jolla, CA, USA) with a group allocation of 1:1. The numbers were subsequently placed into opaque sealed envelopes which were opened 15 min before the patient‘s arrival in the anesthetic room. The technician generating the randomization list was not involved in any other step of the study. Both patients and assessors for the postoperative period were blinded to the group allocation. Blinding of the operators was not possible due to the clearly different designs of the videolaryngoscopes. 

### 2.5. Procedures

Patients in both groups received oral anxiolytic premedication (alprazolam 0.25–0.5 mg) an hour before arriving at the anesthetic room. Standard non-invasive monitoring and oxygen via a face mask was applied. An invasive blood pressure measurement was used where clinically indicated. All patients were placed in the head-elevated laryngoscopy position (HELP) using a modified version of the Oxford pillow. Five minutes of preoxygenation using a tightly sealing mask, 100% oxygen, with continuous positive airway pressure of 5 cmH_2_O was performed in all participants. Induction of general anesthesia was achieved with propofol 2 mg∙kg^−1^, sufentanil 0.15 µg∙kg^−1^, and rocuronium 0.6 mg∙kg^−1^ using an adjusted body weight equation [13]. An appropriate dose of sugammadex was readily available for acute neuromuscular block reversal in case of significant difficulties. An initial attempt on laryngoscopy was performed after confirmation of sufficient muscle paralysis using relaxometry (Datex-Ohmeda Inc., Madison, WI, USA) and a Train of four (TOF) value of zero. Additional doses of propofol were given if the depth of anesthesia was deemed insufficient. A videolaryngoscope was inserted into the mouth in the midline to a depth where both the tip of the epiglottis and vocal cords were seen. After obtaining the optimal view of the vocal cords, the endotracheal tube was inserted into the trachea and controlled ventilation was initiated. Standard cuffed endotracheal tubes (Covidien^TM^, Medtronic Group, Minneapolis, MN, USA) with internal diameters of 7.5 mm for females and 8.5 mm for males were used. The tubes were preloaded into the blade channel in the King Vision (KVL) group whilst in the Glidescope (GS) group, a designated stylet was used. Measurement of the time taken for intubation started at discontinuation of face-mask ventilation and ended when the correct position of the endotracheal tube was confirmed by the first visible etCO_2_ wave on the monitor. A maximum of three laryngoscopy attempts were allowed. Switching the device for one more attempt was allowed if bag-mask ventilation was sufficient; otherwise, the DAS algorithm for unanticipated difficult intubation [14] was strictly followed.

### 2.6. Outcome Measures

The primary outcome of the study was time to tracheal intubation measured by an independent assessor. Measurements were performed from discontinuation of bag-mask ventilation until the confirmation of successful gas exchange on capnography. The second time interval between starting laryngoscopy and insertion of the endotracheal tube was also recorded.

Secondary outcomes included total success rate, the first attempt success rate, number of attempts at tracheal intubation, quantification of the best view achieved on the monitor of the videolaryngoscope using a modified Cormack and Lehane grading [15], number of additional maneuvers, such as external laryngeal pressure (BURP), head elevation or changes in head position, and incidence of significant hypoxemia (oxygen saturation on the pulse oximeter ≤ 85%). The intubation difficulty scale (IDS) [16] was also calculated. Perioperative and postoperative complications including intraoperative trauma to the structures of the oral cavity including the teeth, postoperative sore throat, swallowing difficulty/dysphagia, hoarseness (scale 0–10, 0 = none, 1–3 = mild, 4–6 = moderate, 7–10 = severe) or new onset of cough were recorded at 2 and 24 h postoperatively. 

### 2.7. Statistical Analysis

The sample size for this study was calculated using previously published data on GlideScope videolaryngoscopy in obese patients [6,8,17,18,19,20]. The mean intubation time was 33–69 s in these studies. Based on these results, we selected a mean intubation time as 45 s, with a standard deviation (SD) of 9 s and a meaningful difference of 10% (5 s). Using a power of 80% and α level 5% (type I error) we calculated the minimal sample size as 51 patients in each group. We decided to enrol 55 patients in each branch (total of 110 patients) to compensate for potential drop-outs.

The obtained data were first tested for normal distribution using the Shapiro–Wilk test. According to different data categories and distributions, either the chi-square, Fisher exact test, or Mann–Whitney U tests were used for statistical analysis. *p* values < 0.05 were considered as statistically significant. The linear regression analysis model was applied for evaluation of the relationship between BMI and intubation difficulties, and intubation time. MedCalc Statistical Software version 19.1.5 (MedCalc Software bv, Ostend, Belgium; https://www.medcalc.org; 2020) was used for all comparisons. 

## 3. Results

### 3.1. Demographic Parameters

During the study period, 186 obese patients were screened for enrollment. Following the exclusion of 76 patients who did not meet the inclusion criteria or refused to participate, in total 110 patients were enrolled, 55 in each group (Figure 3). Demographic data and preoperative airway assessment parameters did not show any statistically significant difference between the groups, apart from for mouth opening and neck circumference (Table 1). 

### 3.2. Primary Outcome

The interval from discontinuation of the bag-mask ventilation to the first successful gas exchange as recorded on capnography was significantly shorter in the KVL group (*p* = 0.007), while the difference in time to intubate the patients (endotracheal tube placement) was not statistically significant between the groups (*p* = 0.07) (Table 2) (Figure 4). 

### 3.3. Secondary Outcomes

The total success rate of tracheal intubation was significantly higher in the GS group (*p* = 0.03). Five patients, in whom intubation was not possible using the KingVision laryngoscope, were successfully intubated using the Glidescope. One patient with a massive mucous secretion in the hypopharynx was intubated using the McCoy laryngoscope. Failure to intubate with the KVL was caused by the epiglottis obstructing the view to the cords in three cases, a large tongue with impossible blade insertion in one procedure, fogging of the optics in one patient, and by Cormack–Lehane score 4 in one other patient. Intubation conditions such as a modified Cormack–Lehane score, the total number of intubation attempts, need for any additional maneuvers improving the laryngeal view (application of external laryngeal pressure, change of head position), and the incidence of airway trauma during intubation attempts were similar in both groups (Table 2). Successful intubation on the first attempt was borderline higher in the GS group (*p* = 0.05). The intubation difficulty scale (IDS) did not differ significantly between the groups (*p* = 0.08), however, the incidence of moderate and difficult intubation conditions (IDS ≥ 3) was significantly higher in the KVL group (*p* = 0.015). 

Sub-group analysis of patients with class III obesity (24 patients in each group, BMI ≥ 40 kg∙m^−2^) revealed a shorter time to the first etCO_2_ wave on the monitor in the KVL group 34.0 s [95% CI 32.6–36.7 s] vs. 41.0 s [95% CI 36.7–43.3 s] (*p* = 0.02) but similar time to tracheal tube placement—27.5 [95% CI 24.0–32.5 s] in the GS group vs. 25.0 [95% CI 22.0–28.5 s] (*p* = 0.27). The success rate was higher in the GS group—100% vs. 79.2 in the KVL group (*p* = 0.049).

Linear regression analysis showed that IDS scores in the KVL group were higher in patients with increasing BMI (Figure 5) (*p* = 0.02), while this was not significant in the GS group (*p* = 0.27). There was no correlation between increasing BMI and intubation time in any group (*p* = 0.11). 

There was no statistically significant difference in the decrease of SpO_2_ nor the total number of patients experiencing SpO_2_ lower than 85 % between the groups (Table 2). 

Postoperative complaints were similar in both groups for all parameters studied: hoarseness (*p* = 0.29), pain during swallowing (*p* = 0.75), neck pain (*p* = 0.54), or new onset of cough (*p* = 0.75) (Table 3). No patients experienced major airway trauma such as perforation of the palate, major trauma to the uvula or epiglottis, or persisting hoarseness.

## 4. Discussion

The results of our study showed that intubating times for the non-channeled Glidescope Titanium videolaryngoscope blade as measured by the first effective gas exchange were longer than those of the channeled King Vision blade. However, the times to tracheal tube placement were without significant difference between the groups. This fact was probably caused by the requirement to use and withdraw the preformed stylet in the GS group which prolonged the intubation time by comparison with the KVL group where no stylet was employed. The total success rate was higher with the Glidescope Titanium blade. An important point to note is that all but one patient who experienced failed intubation with the KVL were subsequently successfully intubated using the GS blade. Both methods were associated with a low incidence of periprocedural complications and postoperative complaints. 

Several studies have compared a videolaryngoscope with direct laryngoscopy using a Macintosh blade in the obese population. The older version of the Glidescope videolaryngoscope was compared with the Macintosh laryngoscope in bariatric patients with BMI ≥ 35 kg∙m^−2^ [8]. The authors found a better quality of the vocal cord visualization, decreased IDS scores but longer intubation times in the Glidescope group. Two patients who had failed intubation with the Macintosh blade were intubated with the Glidescope without difficulties. Ander et al. [21] studied the C-MAC videolaryngoscope versus direct laryngoscopy in obese patients and concluded that the incidence of failed tracheal intubation is lower with the videolaryngoscope mainly in male subjects. Three different videolaryngoscopes, including the Glidescope, were compared to the Macintosh blade in obese patients scheduled for bariatric surgery [6]. All videolaryngoscopes provided improved visualization of the laryngeal inlet, with Glidescope and Video-Mac requiring significantly less intubation attempts. The channeled blade of KVL was compared with the C-MAC videolaryngoscope in a cohort of obese patients with predicted difficult airways [22]. Times to visualization of the glottis and tracheal intubation were similar but the KVL group showed a higher incidence of periprocedural complications. However, this study enrolled all subjects with a BMI > 30 kg∙m^−2^. 

Another trial compared six videolaryngoscopes–three non-channeled and three channeled–in a mixed surgical population with a simulated limited mouth opening and neck movements [23]. The performance of the non-channeled blades was found superior to the channeled videolaryngoscopes. The use of videolaryngoscope as a first option has already been adopted by some centers in morbidly obese or in other obese patients with predicted difficult intubation [20,24,25]. 

Our study has several limitations. The anesthesiologists performing the procedures could not be blinded due to the different design of the devices and therefore the possibility of the observer bias cannot be eliminated. The operators were also anesthesiologists experienced in tracheal intubation and the results with less experienced intubators or novices might be different. However, patients with potentially difficult airways are in most institutions intubated by experienced anaesthesiologists. The groups also differed in two of the preoperative airway assessment parameters—mouth opening and neck circumference—which might have affected the results. 

## 5. Conclusions

We conclude that although time to the first capnography is longer in the GS group, and time to tracheal intubation is similar between the KVL and GS Titanium in the obese population with the BMI higher than 35 kg∙m^−2^, the use of GS Titanium videolaryngoscope is associated with fewer failures and with the success in all but one patients where the KVL have previously failed. Therefore, the GS Titanium blade might be preferred in this special population. 

## Figures and Tables

**Figure 1 diagnostics-10-01024-f001:**
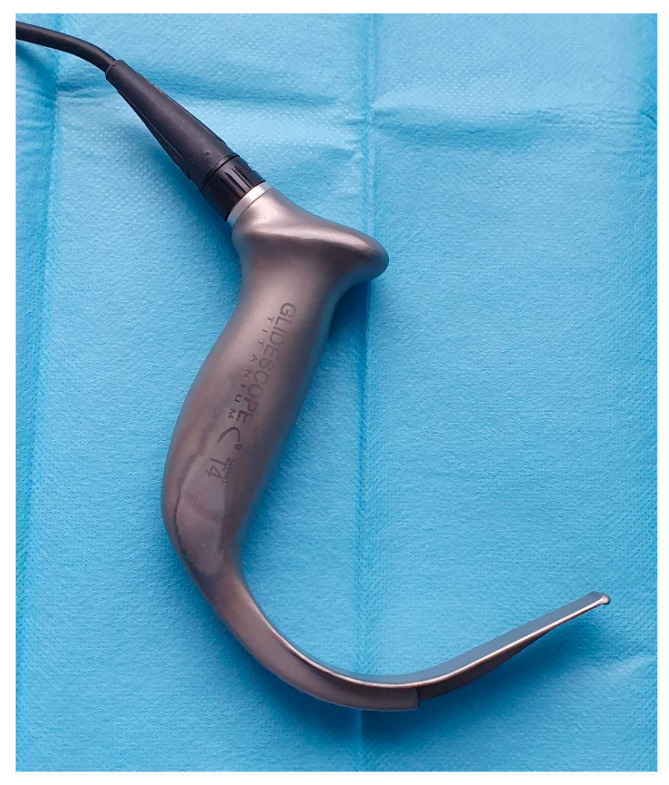
Glidescope Titanium^TM^ videolaryngoscope.

**Figure 2 diagnostics-10-01024-f002:**
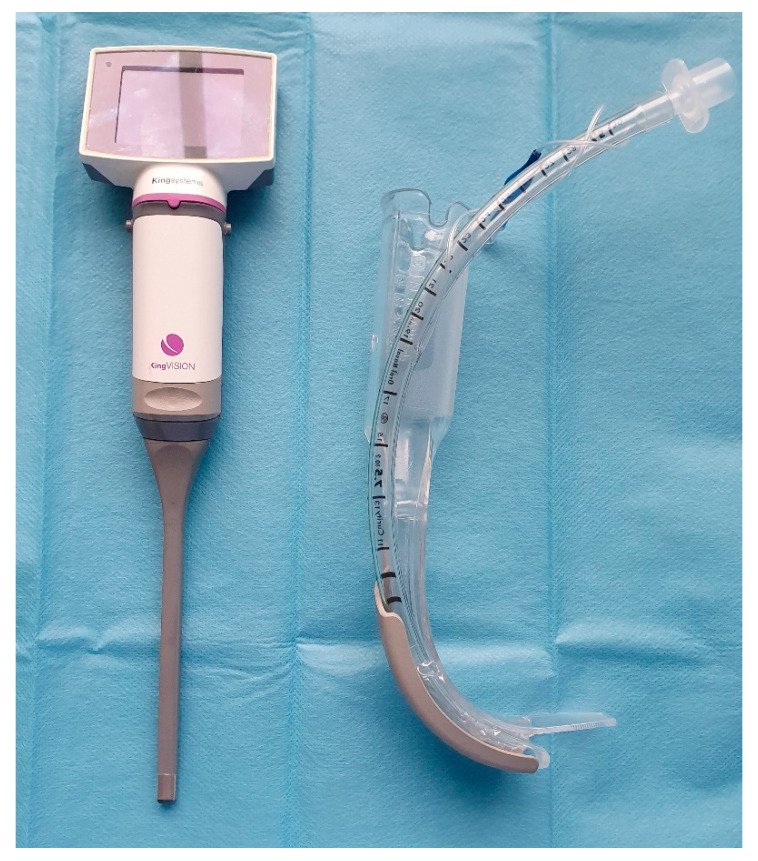
Channeled King Vision^TM^ videolaryngoscope.

**Figure 3 diagnostics-10-01024-f003:**
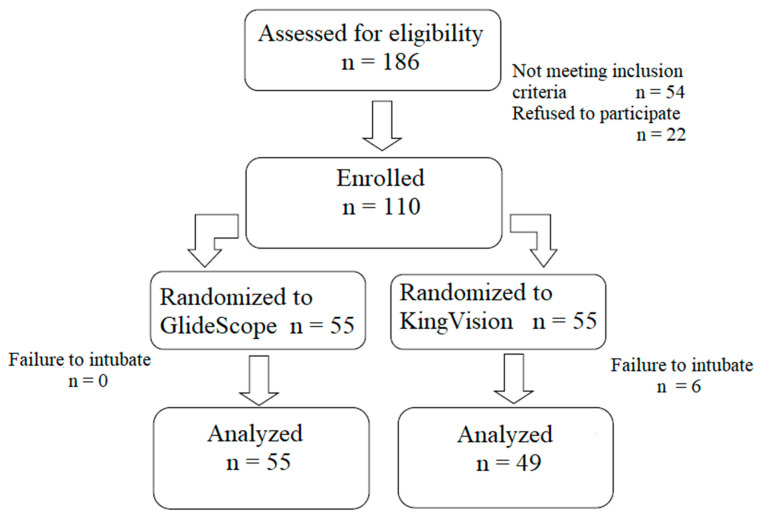
CONSORT 2010 flow diagram of the study.

**Figure 4 diagnostics-10-01024-f004:**
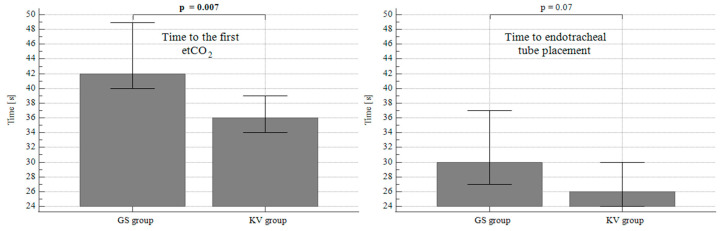
Difference in time intervals to the first capnography and to tracheal tube placement. Data expressed as median, 95 Confidence Interval for median. GS: Glidescope Titanium, KV: King Vision videolaryngoscope.

**Figure 5 diagnostics-10-01024-f005:**
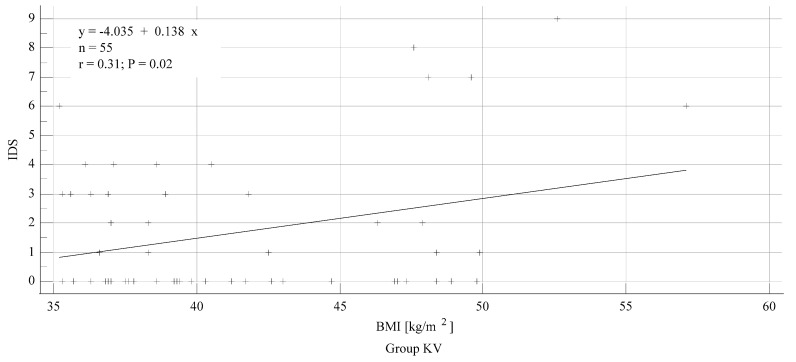
Linear regression model between increasing IDS and BMI in the KVL group.

**Table 1 diagnostics-10-01024-t001:** Demographic parameters and preoperative airway assessment values.

Variable	Glidescope Titanium Group	King Vision Group
	(*n* = 55)	(*n* = 55)
Sex (males/females)	32/23 (58%/42%)	29/26 (53%/47%)
Age (years)	56 [50.5–62] (29–87)	56 [51–61.2] (28–75)
Height (cm)	175 [172–178] (152–200)	172 [170–176] (153–195)
Weight (kg)	120 [119–132] (90–205)	122 [118–130] (89–190)
Body mass index (kg∙m^−2^)	39.5 [37.9–41.1](35.1–63.2)	39.2 [37.6–41.7](35.2–57.1)
American Society of Anesthesiologists	1/34/20/0	0/45/9/1
Status (1/2/3/4)	(2%/52%/36%/0%)	(0%/82%/16%/2%)
Duration of procedure (min)	116 [98–132] (30–300)	90 [89–109] (45–220)
Mallampati score (1/2/3/4)	12/24/16/3	8/28/17/2
	(22%/44%/29%/5%)	(14%/51%/31%/4%)
Mouth opening (cm)	4.5 [4,5] (2–8)	4 [4–4.5] (2–7)
Thyromental distance (cm)	7 [6,7] (3–12)	7 [6.5–7] (4.5–12)
Neck circumference (cm)	49 [47–50] (35–73)	52 [50–55] (36–66)
Limited neck extension (<35°)	19 (35%)	8 (15%)
Sleep apnea syndrome	23 (42%)	21 (38%)

Data expressed as median, [95% confidence interval], (range); total number (%).

**Table 2 diagnostics-10-01024-t002:** Results.

Outcome	Glidescope Titanium	King Vision	*p*
	(*n* = 55)	(*n* = 55)	
Duration of intubation (sec)	42 [40–48.9] (20–147)	36.0 [34–39] (15–98)	0.007 *
Tracheal tube placement (sec)	30.0 [27–37] (12–132)	26.0 [24–30] (10–83)	0.07
Total success rate (n, %)	55 (100%)	49 (89.1%)	0.03 *
First attempt success (n, %)	49 (89.1%)	40 (72.7%)	0.05
Number of attempts 1/2/3 (n)	49/6/0	40/12/3	0.05
Cormack–Lehane grade 1/2/3/4 (n)	39/12/4/0	35/16/3/1	0.57
Intubation Difficulty Scale	35/16/3/1	0 [0–1.2] (0–9)	0.27
Additional maneuvers (n, %)	12 (21.8%)	20 (36.4%)	0.17
Trauma during intubation (n, %)	1 (1.8%)	7 (12.7%)	0.06
Lowest SpO_2_ during attempts (%)	97 [93.4–97] (55–100)	96 [93.5–96.3] (71–100)	0.18

Data expressed as median, [95% CI], (range). * statistically significant

**Table 3 diagnostics-10-01024-t003:** Postoperative complaints.

Outcome	Glidescope Titanium	King Vision	*p*
	(*n* = 55)	(*n* = 55)	
Sore throat (n); none/mild/moderate/severe	21/31/2/1	16/32/6/1	0.54
Hoarseness (n); none/mild/moderate/severe	44/7/4/0	41/11/3/0	0.29
Pain on swallowing (n); none/mild/moderate/severe	25/27/3/0	29/23/2/1	0.75
Cough (n, %)	5 (9.1%)	6 (10.9%)	0.75
Other complaints (n, %)	1 (1.8%)	5 (9.1%)	0.09

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
