# Peer review of "A Randomized Comparison of Non-Channeled GlidescopeTM Titanium Versus Channeled KingVisionTM Videolaryngoscope for Orotracheal Intubation in Obese Patients with BMI > 35 kg·m−2"

_diagnostics, 2020, doi:10.3390/diagnostics10121024_

Round 1
Reviewer 1 Report
This study is well-designed and the manuscript is well-written. I just have one minor suggesionts. The effect of different anesthesiologists may be taken into analysis.
Author Response
This study is well-designed and the manuscript is well-written. I just have one minor suggesionts. The effect of different anesthesiologists may be taken into analysis.
Response: Description of seniority and training of the anesthesiologists performing tracheal intubation in both groups was added to the „Methods“ section of the manuscript.
Five investigators performed all tracheal intubations of the study patients. They were all board-certified anesthesiologists, with at least 8 years of work experience. None of the study devices had been a routine intubation tool in our institution before the start of the trial. The investigators underwent complex training before the initiation of the study. The training included a video demonstration of both devices, intubating manikins with both devices, and at least 20 intubations on non-study patients with each videolaryngoscope prior to enrollment of the first patient.
Reviewer 2 Report
Well designed, conducted and reported prospective randomized single centre study comparing Glidscope and King Vision video laryngoscopes
Abstract: should mention study design
and conclusions should mention as stated in the conclusions of the main text:"All but one patient where intubation failed with the KVL were subsequently successfully intubated using the GS."
Figure 1 and 2 should be optimised to depict only the airway management tool. Especially on figure 1 the background is unnecessarily busy and should be empty and mono coloured as it Figure 2. In Figure 2 the tip of an endotracheal tube left lower corner should not be depicted
Methods: Simplify "patient information pack", do you mean study information and consent form?
Line 91-95 Define the investigators: Are they anaesthesiologists? Which training level? How many years of work experience? Any prior use of Glidescope or King Vision laryngoscope?
Line 108- Consider changing ASA monitoring to non invasive monitoring. Were arterial cannulas inserted? If you deem mentioning ASA monitoring necessary expand on it in order to enable the general reader to understand what it entails
160, 272- Employees are employed by an employer. Devices or tools are used
177- Consider sex instead of gender
273-275- Important statement, should be part of conclusions. See my suggestion for conclusions of abstract
298- here you state for the first time that investigators were anaesthesiologists. Please define the investigators clearly in the methods
Conclusions in the main text should be identical with conclusions of the abstract, amend accordingly
Author Response
Well designed, conducted and reported prospective randomized single centre study comparing Glidscope and King Vision video laryngoscopes
Abstract: should mention study design
Response: We mentioned the design in the „Abstract“ - 110 patients with a body mass index > 35 kg.m-2 were prospectively randomized to tracheal intubation using non-channeled Glidescope Titanium or channeled King Vision videolaryngoscope.
and conclusions should mention as stated in the conclusions of the main text:"All but one patient where intubation failed with the KVL were subsequently successfully intubated using the GS."
Response: Following statement has been added to the “Abstract”. Although tracheal intubation with King Vision showed shorter time to the first breath, total success was higher in the Glidescope group, and all but one patients where intubation failed with the KingVision were subsequently intubated with the Glidescope.
Figure 1 and 2 should be optimised to depict only the airway management tool. Especially on figure 1 the background is unnecessarily busy and should be empty and mono coloured as it Figure 2. In Figure 2 the tip of an endotracheal tube left lower corner should not be depicted
Response: Both Figures have been replaced.
Methods: Simplify "patient information pack", do you mean study information and consent form?
Response: it has been simplified: „participant information sheet“
„The study coordinators described the design of the trial, responded to questions, provided participant information sheet, and informed consent form.“
Line 91-95 Define the investigators: Are they anaesthesiologists? Which training level? How many years of work experience? Any prior use of Glidescope or King Vision laryngoscope?
Response: The investigators an their training was described in the „Methods“ section –
Five investigators performed all tracheal intubations of the study patients. They were all board-certified anesthesiologists, with at least 8 years of work experience. None of the study devices had been a routine intubation tool in our institution before the start of the trial. The investigators underwent complex training before the initiation of the study. The training included a video demonstration of both devices, intubating manikins with both devices, and at least 20 intubations on non-study patients with each videolaryngoscope prior to enrollment of the first patient.
Line 108- Consider changing ASA monitoring to non invasive monitoring. Were arterial cannulas inserted? If you deem mentioning ASA monitoring necessary expand on it in order to enable the general reader to understand what it entails
Response: We agree, we have changed the text accordingly.
„Standard non-invasive monitoring and oxygen via a face mask was applied. An invasive blood pressure measurement was used where clinically indicated.“
160, 272- Employees are employed by an employer. Devices or tools are used
Response: We agree that the word „employed“ is not usual in American English in this context. We have replaced this with the word „used“ as recommended by the reviewer.
„MedCalc Statistical Software version 19.1.5 (MedCalc Software bv, Ostend, Belgium; https://www.medcalc.org; 2020) was used for all comparisons.“
177- Consider sex instead of gender
Response: The word „gender“ has been replaced by „sex“
273-275- Important statement, should be part of conclusions. See my suggestion for conclusions of abstract
Response: We have changed the „Conclusions“ accordingly –
„We conclude that although time to the first capnography is longer in the GS group, and time to tracheal intubation is similar between the KVL and GS Titanium in the obese population with the BMI higher than 35 kg-m-2, the use of GS Titanium videolaryngoscope is associated with fewer failures and with the success in all but one patients where the KVL have previously failed. Therefore the GS Titanium blade might be preferred in this special population.“
298- here you state for the first time that investigators were anaesthesiologists. Please define the investigators clearly in the methods
Response: Description of the investigators has been added to the „Methods“ -
Five investigators performed all tracheal intubations of the study patients. They were all board-certified anesthesiologists, with at least 8 years of work experience. None of the study devices had been a routine intubation tool in our institution before the start of the trial. The investigators underwent complex training before the initiation of the study. The training included a video demonstration of both devices, intubating manikins with both devices, and at least 20 intubations on non-study patients with each videolaryngoscope prior to enrollment of the first patient.
Conclusions in the main text should be identical with conclusions of the abstract, amend accordingly
Response: We have changed the content of the „Conclusions“ accordingly to match with the „Abstract“:
„We conclude that although time to the first capnography is longer in the GS group, and time to tracheal intubation is similar between the KVL and GS Titanium in the obese population with the BMI higher than 35 kg-m-2, the use of GS Titanium videolaryngoscope is associated with fewer failures and with the success in all but one patients where the KVL have previously failed. Therefore the GS Titanium blade might be preferred in this special population.“